# Impact of depression on personal hygiene practices- A cross-sectional study among university students in Bangladesh

Fouzia Akter◉*, Akibul Islam Chowdhury, Md. Nawal Sarwer◉

Nutrition and Food Engineering Department, Daffodil International University, Dhaka, Bangladesh

* fouzia@daffodilvarsity.edu.bd

## Abstract

### Background

This study explores the relationship between depression and personal hygiene practices among university students in Bangladesh.

### Methods

A cross-sectional online survey was conducted, utilizing an 18-item Personal Hygiene Practice Questionnaire (PHPQ) and the Center for Epidemiologic Studies Depression Scale (CES-D) to assess hygiene behaviors and depression risk among 1,913 undergraduate students in Dhaka, Bangladesh. Data were analyzed using chi-square test and ordered logistic regression. The PHPQ was validated through item analysis, internal consistency, construct validity and reliability tests.

### Results

A high prevalence of depression risk was revealed with 79.9% of females and 73.9% of males. Females demonstrated superior hygiene practices, with 90.1% classified as having good hygiene compared to 75.0% of males. Accommodation type significantly influenced both depression and hygiene, as students living in privately managed housing exhibited better hygiene practices (88.6% good hygiene) and lower depression risk (73.2%) compared to those living at home (79.2%) or in university housing (78.7%). Ordered logistic regression analysis indicated that students at risk of depression had 65% lower odds of maintaining better hygiene practices (OR = 0.36, p < 0.001), and male students were 68% less likely to have higher hygiene scores than females (OR = 0.32, p < 0.001). The Exploratory Factor Analysis and Cronbach's alpha confirmed the reliability (α = 0.83) and strong internal consistency of PHPQ-18 scale.

**Data availability statement:** All relevant data are within the manuscript and its Supporting Information files.

**Funding:** The author(s) received no specific funding for this work.

**Competing interests:** The authors have declared that no competing interests exist.

## Conclusion

These findings underscore the need for targeted interventions in university settings to address mental health and hygiene education. Further research should explore socio-economic and cultural factors influencing these relationships.

## Introduction

Depression is a common mental health disease that has a substantial influence on people's daily life, including their personal hygiene habits. As a common yet often under-recognized condition, depression affects millions worldwide, with a notable prevalence among university students [1]. This demographic is particularly vulnerable due to the unique stressors associated with academic life, including academic performance, studying in the English language, heavy lecture schedule, pressure to succeed, future planning, and social challenges [2]. Other factors associated with mental illness are demographic factor, including gender, residence, relationship status, socioeconomic status, loneliness, personal autonomy, family and peer pressure [2–4].

Personal hygiene is crucial for maintaining good health, preventing infectious diseases, and promoting psychological and social well-being [5]. Individuals experiencing depression frequently struggle with daily self-care routines due to symptoms like low motivation, fatigue, and cognitive difficulties. Neglecting hygiene can further contribute to poor physical health, social isolation, and decreased self-esteem, potentially exacerbating depressive symptoms in a detrimental cycle of declining mental and physical well-being [6,7]. The relationship between personal hygiene and depression needs a comprehensive understanding as it is not well established.

At the University level, students have experienced a transition to adulthood. They have moved away from their family into new places and cope with new environment [4] which might lead to both positive and negative changes in their lifestyles [8]. Some students may adopt a healthier lifestyle while others may struggle with their new environment and academic life which may have an impact on their personal health related activities.

Previous studies highlighted that depression among university students in Bangladesh is a significant concern, with prevalence rates ranging from 28.7% to 47.3% [9]. Several factors contribute to depression, including years of study, stressful life events, suicidal attempts, inadequate monthly allowance, substance use, physical and psychological illness, and excessive social media use [10]. The prevalence of depression among university students is rising, yet studies exploring its impact on personal hygiene remain limited.

The relationship between personal hygiene and depression is multifaceted. Previous studies at university settings have shown association between depression and unhealthy lifestyles along with disruption in daily routine activities [8,11]. Depression can lead to a lack of motivation and energy, making routine care activities feel overwhelming. Some studies showed hygiene related practices at school settings may

influence absenteeism and poor academic performance [12] which may also lead to depression. Moreover, poor personal hygiene may have an impact on poor individual health and social responsibilities. Students with poor personal hygiene may face social stigmatization leading to isolation and depression. Despite these concerns, limited empirical research has examined the relationship between depression and hygiene behaviors among university students, highlighting a significant gap in literature. Therefore, addressing personal hygiene within the context of mental health is vital.

To address this gap, our study explores the association between depression and personal hygiene practices among university students in Bangladesh using a novel 18-item Personal Hygiene Practice Questionnaire (PHPQ).

## Methodology

### Study design and location

We conducted this cross-sectional study among university students using an online survey from Dhaka city due to its significant concentration of universities. Dhaka has the highest number of universities in Bangladesh with a university present in every 5.38 square kilometers. Due to this Dhaka is known as a central hub for higher education in the country [13].

### Sampling and data collection

We used a convenience sampling method to recruit participants for the study. A structured questionnaire was distributed via a Google Form to ensure accessibility for participants across various locations with the study nature, purpose and eligibility inclusion criteria for participation. The inclusion criteria required for the participants to be 1) a resident in Dhaka, 2) studying either public or private university, 3) studying in first/second/third/fourth year, 4) studying health or non-health background subject majors, 5) have access to computer or mobile with internet connection, 6) able to understand English and 7) written consent for participation. This online method of data collection enhances self-disclosure on sensitive topics [14]. Participation was voluntary, students who were interested accessed and completed the questionnaire. We shared invitations for participation in this study across both public and private university networks, student groups, and social media platforms. Recruitment of participants for data collection started on September 8, 2024, and ended on December 31, 2024. The survey link was shared online on different platforms. Initially, 2,030 participants provided written informed consent online. After applying exclusion criteria for missing values and inconsistencies, 1913 respondents completed the entire survey, generating a response rate of 94.23%.

As we used a convenience sampling approach, this might introduce sampling bias since participation depended on students' willingness to respond. Although we attempted to reduce this by distributing the survey across public and private universities, male and female student groups, and both health- and non-health-related majors, the final sample may not fully represent all university students in Dhaka.

### Study measures

The socio-demographic section contained data on age, gender, type of university, students' field of study and their level of study, place of residence, parental education, and family income in Bangladeshi currency (BDT). We used the quartile method based on participant's self-reported monthly family income and divided the continuous income variable into four quartiles, where each quartile represents 25% of the sample. The cut-off value of each quartile was determined by Stata based on the distribution of family income variable in our dataset.

### The center for epidemiologic studies depression scale (CES-D)

In this study, we used CES-D scale to assess depression among university students in Dhaka, Bangladesh, as it has been validated in various population including young adults and in low middle income countries. It is comprised of 20 self-reported items that measure different dimensions of depression. Respondents rate the frequency of each symptom over

the past week on a four-point Likert scale, ranging from 0 (rarely or none of the time, less than 1 day) to 3 (most or all the time, 5–7 days). The total score range is 0–60. A score of 16 and above was used to define a case of likely depression or at risk of depression and a score less than 16 was defined as not at risk of depression [15].

**Personal hygiene practice questionnaire**

We developed a new questionnaire (Fig 1) to assess personal hygiene practices among university students, guided by discussions with faculty members from the Departments of Nutrition and Food Engineering, Pharmacy, and Public Health. Expert review led to the refinement of an initial 21-item questionnaire, resulting in an 18-item final version that encompasses various dimensions of personal hygiene, including hand hygiene and personal cleanliness. After developing the questionnaire, a pilot survey was conducted with 44 students to check whether they could easily understand the questions

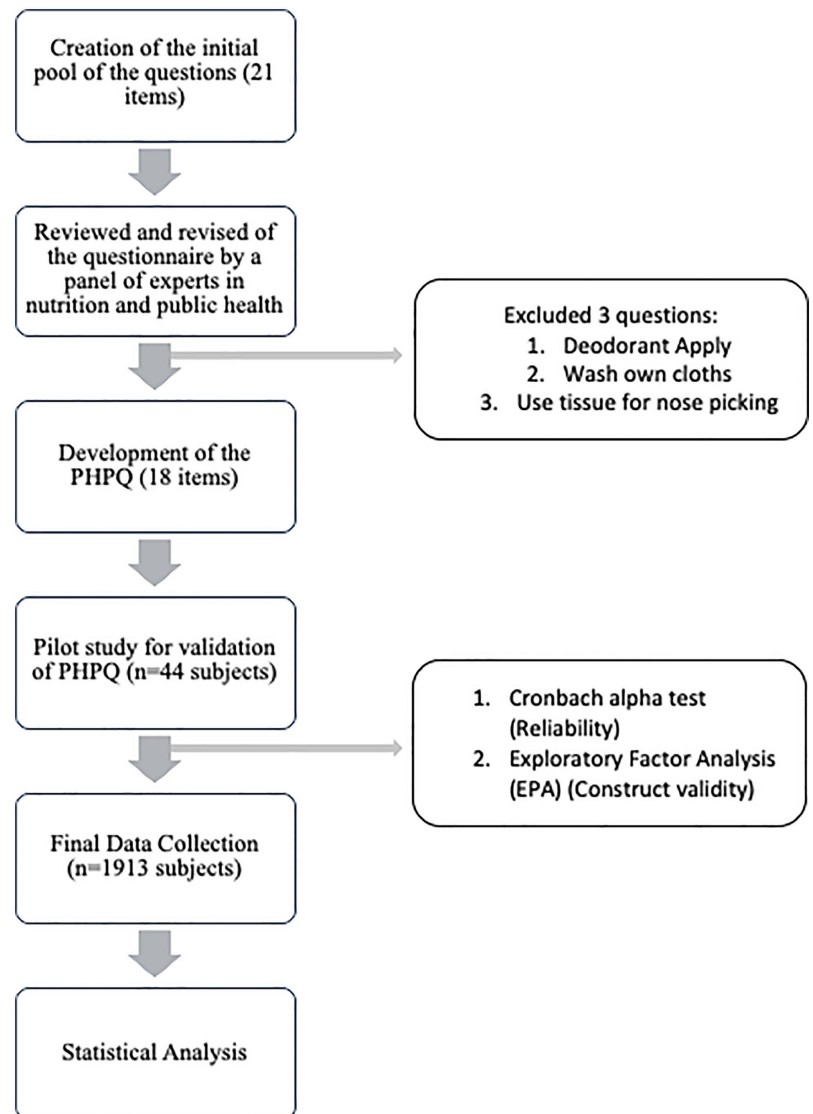

**Fig 1. Flowchart of PHPQ development and validation process.**

and response options. 93.18% of students didn't find any difficulties in understanding the question and 86.36% of students had no issues with the options of the questions. We also tested the internal reliability of the questionnaire using Cronbach's alpha and got a value of 0.77, indicating acceptable reliability for measuring personal hygiene practices. After considering students feedback, we have included both three-point and four-point Likert scales response format to quantify the frequency of hygiene practice. Items on a three-point scale were scored as (i) "Always" = 2, "Sometimes" = 1, "Never" = 0; (ii) "Daily" = 2, "Weekly" = 1, "Monthly" = 0. For the four-point scale, responses were coded as "Once a week" = 2, "Once in 15 days" = 1, "Once a month" = 1, and "Once more than one month duration" = 0. The total hygiene score ranged from 0 to 36, with higher scores indicating better hygiene practices. Participants were classified into three hygiene categories: poor (0–17), moderate (18–26), and good (27–36).

### Statistical analysis

Continuous socio-demographic variables using mean, standard deviation and range and categorical variables using frequency and percentage were measured. Chi-square test ($\chi^2$) was used to determine any statistically significant association between socio-demographic variables, depression and personal hygiene scores. Cronbach's alpha for reliability test and Exploratory Factor Analysis (EFA) were done to test the validity of our newly developed personal hygiene practice questionnaire. After that an ordered logistic regression analysis along with model assumptions test (S1 Table) was used to analyze how depression and socio-demographic factors affect personal hygiene practice. Potential confounders, including age, gender, university type, subject major, year of study, residence type, parental education, and family income quartile were adjusted in the regression analysis. All statistical analyses were done using Stata 18. A p-value less than 5% was considered as statistically significant.

### Ethical consideration

The study was conducted in accordance with the Declaration of Helsinki. We obtained written informed consent from all participants provided through a separate page attached before the questionnaire. Participants were informed about nature and purpose of the study, and their right to withdraw from the study any time they want. Ethical approval for this study was provided by the Research Ethical Committee of Daffodil International University (DIU) and the approval number is FAHSREC/DIU/2024/SMIG-18.

## Results

### Participants' characteristics

The total number of participants in this study was 1,913 with a mean age of 21.99 years (SD = 1.61) and an age range of 18–33 years (Table 1). Percentage of males (52.17%) are slightly higher than females (47.83%). The distribution of students between public (50.65%) and private universities (49.35%) was nearly equal. Among field of study, engineering students comprised the largest group (39.57%), followed by those from other disciplines (38.63%), life sciences (13.80%), and medical fields (8.0%).

All continuous variables are expressed in mean and standard deviation.

Regarding study level, second-year students represented the largest proportion (40.72%), followed by first-year (23.58%), third-year (22.69%), and fourth-year students (13.02%). Place of residence varied among students, with 38.26% living at home, 27.71% residing in university-provided housing, and 34.03% in private accommodation.

Data on parental education levels presented that 37.22% of students' mothers and 21.59% of fathers had attained secondary education and 42.45% of fathers and 22.79% of mothers had graduate or postgraduate degrees. We categorized the family income of our participants into income quartiles, where 27.50% in the lowest quartile (Q1), 31.00% in the second quartile (Q2), 17.98% in the third quartile (Q3), and 23.52% in the highest quartile (Q4).

**Table 1. Socio-demographic information of the participants.**

| Variables | Category | n (%) |
|---|---|---|
| **Age (in years)** | Mean ± SD | 21.99 ± 1.61 |
| | Range | 18-33 |
| **Gender** | Female | 915 (47.83) |
| | Male | 998 (52.17) |
| **Type of University** | Public | 969 (50.65) |
| | Private | 944 (49.35) |
| **Field of Study** | Medical | 153 (8.0) |
| | Life Sciences | 264 (13.80) |
| | Engineering | 757 (39.57) |
| | Others | 739 (38.63) |
| **Study Level** | First Year | 451 (23.58) |
| | Second Year | 779 (40.72) |
| | Third Year | 434 (22.69) |
| | Fourth Year | 249 (13.02) |
| **Accommodation** | Home | 732 (38.26) |
| | Provided by the University | 530 (27.71) |
| | Privately Managed | 651 (34.03) |
| **Education of Mother** | Illiterate | 40 (2.09) |
| | Primary | 189 (9.88) |
| | Secondary | 712 (37.22) |
| | Undergraduate | 536 (28.02) |
| | Graduate and post-graduate | 436 (22.79) |
| **Education of Father** | Illiterate | 41 (2.19) |
| | Primary | 120 (6.27) |
| | Secondary | 413 (21.59) |
| | Undergraduate | 527 (27.55) |
| | Graduate and post-graduate | 812 (42.45) |
| **Family Income** | Lowest (Q1) | 526 (27.50) |
| | Second (Q2) | 593 (31.00) |
| | Third (Q3) | 344 (17.98) |
| | Highest (Q4) | 450 (23.52) |

All categorical variables are expressed in percentage (%).

### Distribution of socio-demographic characteristics according to depression and personal hygiene practice

The relationship between socio-demographic factors and both depression risk (CES-D score categories) and personal hygiene practice categories among university students are presented in Table 2. Study findings show that gender significantly influences both depression risk and personal hygiene practices. A higher proportion of females (79.9%) were at risk of depression compared to males (73.9%) ($\chi^2 = 9.77$, p = 0.002). Females demonstrated better personal hygiene practices, with 90.1% classified as having good hygiene compared to 75.0% of males ($\chi^2 = 74.73$, p < 0.001).

Accommodation type was significantly associated with depression and personal hygiene. Students living in privately managed accommodations had a lower risk of depression (73.2%) compared to those living at home (79.2%) or in university-provided housing (78.7%) ($\chi^2 = 8.26$, p = 0.016). They also showed better hygiene practices, with 88.6%

**Table 2. Association of Socio-Demographic variables with CES-D Category and Personal Hygiene Practice Category.**

| Socio-Demographic Variables | CES-D Category | | | χ2 | p-value | Personal Hygiene Practice | | | | χ2 | p-value |
|---|---|---|---|---|---|---|---|---|---|---|---|
| | Not at risk (n, %) | At risk (n, %) | Total (n, %) | | | Poor | Moderate | Good | Total (n, %) | | |
| **Gender** | | | | | | | | | | | |
| Female | 184 (20.1) | 731 (79.9) | 915 (100) | 9.77 | **0.002** | 4 (0.4) | 87 (9.5) | 824 (90.1) | 915 (100) | 74.73 | **<0.001** |
| Male | 261 (26.1) | 737 (73.9) | 998 (100) | | | 16 (1.6) | 234 (23.4) | 748 (75.0) | 998 (100) | | |
| **University Type** | | | | | | | | | | | |
| Public | 230 (23.7) | 739 (76.3) | 969 (100) | 0.25 | 0.619 | 7 (0.7) | 175 (18.1) | 787 (81.2) | 969 (100) | 4.10 | 0.129 |
| Private | 215 (22.8) | 729 (77.2) | 944 (100) | | | 13 (1.4) | 146 (15.5) | 785 (83.1) | 944 (100) | | |
| **Study Major** | | | | | | | | | | | |
| Medical | 45 (29.4) | 108 (70.6) | 153 (100) | 3.78 | 0.286 | 2 (1.3) | 26 (17.0) | 125 (81.7) | 153 (100) | 6.83 | 0.337 |
| Life Sciences | 57 (21.6) | 207 (78.4) | 264 (100) | | | 2 (0.8) | 37 (14.0) | 225 (85.2) | 264 (100) | | |
| Engineering | 175 (23.1) | 582 (76.9) | 757 (100) | | | 5 (0.7) | 142 (18.8) | 610 (80.5) | 757 (100) | | |
| Others | 168 (22.7) | 571 (77.3) | 739 (100) | | | 11 (1.5) | 116 (15.7) | 612 (82.8) | 739 (100) | | |
| **Study Level** | | | | | | | | | | | |
| First Year | 108 (23.9) | 343 (76.1) | 451 (100) | 1.36 | 0.714 | 3 (0.7) | 58 (12.9) | 390 (86.5) | 451 (100) | 12.43 | **0.053** |
| Second Year | 171 (22.0) | 608 (78.0) | 779 (100) | | | 13 (1.7) | 135 (17.3) | 631 (81.0) | 779 (100) | | |
| Third Year | 104 (24.0) | 330 (76.0) | 434 (100) | | | 3 (0.7) | 79 (18.2) | 352 (81.1) | 434 (100) | | |
| Fourth Year | 62 (24.9) | 187 (75.1) | 249 (100) | | | 1 (0.4) | 49 (19.7) | 199 (79.9) | 249 (100) | | |
| **Accommodation** | | | | | | | | | | | |
| Home | 110 (20.8) | 420 (79.2) | 530 (100) | 8.26 | **0.016** | 5 (0.9) | 115 (21.7) | 410 (77.4) | 530 (100) | 36.82 | **<0.001** |
| Provided by the University | 139 (21.3) | 512 (78.7) | 651 (100) | | | 7 (1.1) | 131 (20.1) | 513 (78.8) | 651 (100) | | |
| Privately Managed | 196 (26.8) | 536 (73.2) | 732 (100) | | | 8 (1.1) | 75 (10.2) | 649 (88.6) | 732 (100) | | |
| **Education of Mother** | | | | | | | | | | | |
| Illiterate | 8 (20.0) | 32 (80.0) | 40 (100) | 3.34 | 0.503 | 0 (0.0) | 14.(35.0) | 26 (65.0) | 40 (100) | 22.48 | **0.004** |
| Primary | 37 (19.6) | 152 (80.4) | 189 (100) | | | 2 (1.1) | 46 (24.3) | 141 (74.6) | 189 (100) | | |
| Secondary | 177 (24.9) | 535 (75.1) | 712 (100) | | | 8 (1.1) | 112 (15.7) | 592 (83.2) | 712 (100) | | |
| Undergraduate | 128 (23.9) | 408 (76.1) | 536 (100) | | | 3 (0.6) | 86 (16.0) | 447 (83.4) | 536 (100) | | |
| Graduate and post-graduate | 95 (21.8) | 341 (78.2) | 436 (100) | | | 7 (1.6) | 63 (14.4) | 366 (84.0) | 436 (100) | | |
| **Education of Father** | | | | | | | | | | | |
| Illiterate | 9 (21.9) | 32 (78.1) | 41 (100) | 0.27 | 0.991 | 0 (0.0) | 11 (26.8) | 30 (73.2) | 41 (100) | 16.35 | **0.038** |
| Primary | 29 (24.2) | 91 (75.8) | 120 (100) | | | 1 (0.8) | 33 (27.5) | 86 (71.7) | 120 (100) | | |
| Secondary | 97 (23.5) | 316 (76.5) | 413 (100) | | | 4 (1.0) | 65 (15.7) | 344 (83.3) | 413 (100) | | |
| Undergraduate | 119 (22.6) | 408 (77.4) | 527 (100) | | | 5 (0.9) | 75 (14.2) | 447 (84.8) | 527 (100) | | |
| Graduate and post-graduate | 191 (23.5) | 621 (76.5) | 812 (100) | | | 10 (1.2) | 137 (16.9) | 665 (81.9) | 812 (100) | | |
| **Family Income** | | | | | | | | | | | |
| Lowest (Q1) | 116 (22.1) | 410 (77.9) | 526 (100) | 3.93 | 0.270 | 5 (0.9) | 112 (21.3) | 409 (77.8) | 526 (100) | 13.13 | **0.040** |
| Second (Q2) | 133 (22.4) | 460 (77.6) | 593 (100) | | | 5 (0.8) | 91 (15.4) | 497 (83.8) | 593 (100) | | |
| Third (Q3) | 94 (27.3) | 250 (72.7) | 344 (100) | | | 3 (0.9) | 46 (13.4) | 295 (85.8) | 344 (100) | | |
| Highest (Q4) | 102 (22.7) | 348 (77.3) | 450 (100) | | | 7 (1.6) | 72 (16.0) | 371 (82.4) | 450 (100) | | |

Other socio-demographic variables, such as university type, subject major, and level of study did not show statistically significant associations with depression risk or personal hygiene practices.

categorized as having good hygiene, compared to 77.4% and 78.8% for home and university-provided housing, respectively ($\chi^2 = 36.82$, p < 0.001).

Parental education level also influenced personal hygiene, students whose mothers had lower education levels exhibiting poorer hygiene ($\chi^2 = 22.48$, p = 0.004). Similarly, father's education showed a significant association with hygiene practices ($\chi^2 = 16.35$, p = 0.038), though it did not significantly impact depression risk (p = 0.991).

Family income was also significantly related to personal hygiene practices ($\chi^2 = 13.13$, p = 0.04). Students from the lowest income quartile (Q1) had poorer hygiene practices than those from higher income groups, though income did not show a significant relationship with depression risk (p = 0.27).

### Depression characteristics by personal hygiene practices

Table 3 demonstrated the association between depression (not at risk and at risk) and personal hygiene practice (poor, moderate and good). A statistically significant association between depression and personal hygiene practice ($\chi^2 = 31.45$, p < 0.001) were found. The majority (91.0%) of the students who were not at risk of depression had good hygiene practices, 8.8% had moderate and only 0.2% had poor hygiene practices. On the other hand, 79.5% of students who were at risk of depression had good hygiene practices, 19.2% had moderate and 1.3% had poor hygiene practices. The chi-square results show that students at risk of depression were less likely to maintain good hygiene practices and more likely to fall into moderate or poor hygiene categories compared to those who were not at risk of depression.

### Association of depression and socio-demographic characteristics with personal hygiene practices

An ordered logistic regression analysis was conducted to explore the impact of depression and socio-demographic factors on personal hygiene practices among university students (**Table 4**). This model was adjusted for key socio-demographic and socioeconomic variables, including age, gender, university type, subject major, year of study, residence type, parental education, and family income quartile, to control the effect of potential confounding variables. Students at risk of depression had 65% lower odds of better hygiene practices compared to those not at risk (OR = 0.36, p < 0.001). Male students were 68% less likely to have higher hygiene scores than females (OR = 0.32, p < 0.001). Accommodation type also showed a significant association with personal hygiene practices. Students living in privately managed accommodations had nearly 2 times higher odds of better hygiene practices compared to those living at home (OR = 1.99, 95% CI: 1.42–2.80, p < 0.001). Second-year students were 34% less likely to practice better hygiene compared to first-year students (OR = 0.66, 95% CI: 0.47–0.94, p = 0.022). However, no significant differences were observed for third- and fourth-year students.

Although higher maternal education showed a positive association with better hygiene practices, the results were not statistically significant. For example, students whose mothers had a postgraduate education had 2.28 times higher odds of better hygiene practices than students whose mothers were illiterate, but the effect did not reach significance (OR = 2.28, 95% CI: 0.88–5.86, p = 0.089). Paternal education, university type, subject major, age, and family income did not show statistically significant associations with personal hygiene practices, as their 95% confidence intervals included 1.

By adjusting for potential confounders, we strengthened the model's reliability. The model was statistically significant ($\chi^2 = 172.78$, p < 0.001), and explained approximately 8.9% of the variance in personal hygiene practices (Pseudo

**Table 3. Association between Depression and Personal Hygiene Practice among University Students.**

| CES-D Score Category | Poor (n,%) | Moderate (n,%) | Good (n,%) | Total (n,%) | $\chi^2$ | df | p-value |
|---|---|---|---|---|---|---|---|
| Not at risk | 1 (0.2) | 39 (8.8) | 405 (91.0) | 445 (100) | 31.45 | 2 | **<0.001** |
| At risk | 19 (1.3) | 282 (19.2) | 1,167 (79.5) | 1,468 (100) | | | |
| Total | 20 (1.0) | 321 (16.8) | 1,572 (82.2) | 1,913 (100) | | | |

**Table 4. Ordered Logistic Regression Analysis of Variables Influencing Personal Hygiene Practice.**

| Variable | Odds Ratio | Std. Error | z-value | p-value | 95% CI |
|---|---|---|---|---|---|
| **Depression** | | | | | |
| Not at risk | Reference group | | | | |
| At risk | 0.36 | 0.07 | −5.65 | **<0.001** | [0.25, 0.51] |
| **Age** | 0.98 | 0.05 | −0.45 | 0.655 | [0.89, 1.08] |
| **Gender** | | | | | |
| Female | Reference group | | | | |
| Male | 0.32 | 0.05 | −7.90 | **<0.001** | [0.24, 0.42] |
| **University type** | | | | | |
| Public | Reference group | | | | |
| Private | 0.99 | 0.14 | −0.06 | 0.955 | [0.75, 1.31] |
| **Subject Major** | | | | | |
| Medical | Reference group | | | | |
| Life Sciences | 1.44 | 0.41 | 1.27 | 0.204 | [0.82, 2.53] |
| Engineering | 1.26 | 0.32 | 0.93 | 0.353 | [0.77, 2.06] |
| Other | 1.16 | 0.29 | 0.59 | 0.555 | [0.71, 1.88] |
| **Study Level** | | | | | |
| First year | Reference group | | | | |
| Second Year | 0.66 | 0.12 | −2.30 | **0.022** | [0.47, 0.94] |
| Third Year | 0.79 | 0.17 | −1.09 | 0.276 | [0.52, 1.20] |
| Fourth Year | 0.82 | 0.23 | −0.69 | 0.488 | [0.48, 1.42] |
| **Accommodation** | | | | | |
| Home | Reference group | | | | |
| University-Provided | 1.14 | 0.18 | 0.86 | 0.392 | [0.84, 1.56] |
| Privately Managed | 1.99 | 0.34 | 4.00 | **<0.001** | [1.42, 2.80] |
| **Mother's Education** | | | | | |
| Illiterate | Reference group | | | | |
| Primary | 1.61 | 0.71 | 1.07 | 0.285 | [0.67, 3.84] |
| Secondary | 1.89 | 0.85 | 1.41 | 0.159 | [0.78, 4.56] |
| Undergraduate | 2.15 | 1.00 | 1.64 | 0.101 | [0.86, 5.37] |
| Graduate/Postgraduate | 2.28 | 1.10 | 1.70 | 0.089 | [0.88, 5.86] |
| **Father's Education** | | | | | |
| Illiterate | Reference group | | | | |
| Primary | 0.68 | 0.32 | −0.82 | 0.413 | [0.26, 1.73] |
| Secondary | 1.05 | 0.50 | 0.11 | 0.912 | [0.41, 2.69] |
| Undergraduate | 1.04 | 0.50 | 0.08 | 0.940 | [0.40, 2.68] |
| Graduate/Postgraduate | 0.67 | 0.33 | −0.81 | 0.418 | [0.26, 1.75] |
| **Family Income** | | | | | |
| Lowest Quartile, Q1 | Reference group | | | | |
| Second Quartile, Q2 | 1.19 | 0.20 | 1.05 | 0.294 | [0.86, 1.66] |
| Third Quartile, Q3 | 1.27 | 0.26 | 1.13 | 0.257 | [0.84, 1.91] |
| Highest Quartile, Q4 | 1.09 | 0.20 | 0.46 | 0.646 | [0.76, 1.57] |
| **Cut points** | | | | | |
| Cut 1 | −5.89 | 1.18 | | | |
| Cut 2 | −2.75 | 1.16 | | | |
| **Model Fit** | | | | | |
| Observations | 1913 | | | | |

*(Continued)*

**Table 4.** (Continued)

| Variable | Odds Ratio | Std. Error | z-value | p-value | 95% CI |
|---|---|---|---|---|---|
| Log likelihood | −886.43 | | | | |
| LR chi²(23) | 172.78 | | | | |
| Prob > chi² | **<0.001** | | | | |
| Pseudo R² (%) | 8.90 | | | | |

$R^2 = 0.0888$). The small effect sizes suggest that the additional factors (e.g., awareness on hygiene related diseases, individual belief about cleanliness, availability of hygiene facilities, social and cultural influences etc.) beyond those included in this analysis may contribute to personal hygiene behavior.

### Reliability and validity of the personal hygiene practice questionnaire

A Cronbach's alpha over 0.70 indicates that the data are reliable and consistently measure a construct [16]. In our study the overall Cronbach's alpha for the questionnaire we used to assess personal hygiene practice was 0.83 which means good internal consistency (Table 5). For most of our items, the item-test correlation ranging from 0.23 to 0.63 exceeded 0.40 which indicated that there was a strong association between each item and the overall scale. All item-rest correlations (ranging from 0.16 to 0.56) except the "cutting nails" item had contributions to the personal hygiene practice scale's internal consistency. The average inter-item covariance of 0.06 demonstrated consistency across the 18-item of our questionnaire.

Before conducting inferential statistics (chi-square and regression test), Exploratory Factor Analysis (EFA) was done to test the validity of the personal hygiene questionnaire. The results of the EFA supported the construct validity of the 18-item personal hygiene practice questionnaire (Table 6). The value of the Kaiser– Meyer–Olkin (KMO) test and the

**Table 5.** Internal Reliability of Personal Hygiene Practice Questionnaire.

| Items | Item-test correlation | Item-rest Correlation | Average interitem covariance | alpha (α) |
|---|---|---|---|---|
| Handwash before eating | 0.45 | 0.39 | 0.06 | 0.83 |
| Handwash after eating | 0.48 | 0.42 | 0.06 | 0.83 |
| Handwash with soap after toilet use | 0.61 | 0.52 | 0.06 | 0.82 |
| Handwash after blowing nose | 0.53 | 0.44 | 0.06 | 0.83 |
| Handwash after touching animal | 0.61 | 0.51 | 0.05 | 0.82 |
| Handwash after touching private parts | 0.63 | 0.56 | 0.06 | 0.82 |
| Brushing teeth | 0.48 | 0.43 | 0.06 | 0.83 |
| Bathing | 0.51 | 0.45 | 0.06 | 0.83 |
| Wear clean clothes | 0.58 | 0.51 | 0.06 | 0.82 |
| Change underwear | 0.61 | 0.55 | 0.06 | 0.82 |
| Removing unwanted hair | 0.53 | 0.45 | 0.06 | 0.83 |
| Wash hair | 0.52 | 0.44 | 0.06 | 0.83 |
| Handkerchief/tissue use after blowing nose | 0.62 | 0.54 | 0.06 | 0.82 |
| Cutting nail | 0.23 | 0.16 | 0.06 | 0.84 |
| Changing bedsheet | 0.49 | 0.39 | 0.06 | 0.83 |
| Changing pillow cover | 0.48 | 0.37 | 0.06 | 0.83 |
| Dusting own room | 0.47 | 0.36 | 0.06 | 0.83 |
| Mopping own room | 0.47 | 0.36 | 0.06 | 0.83 |
| Total Scale | | | 0.06 | 0.83 |

**Table 6. Exploratory Factor Analysis of the Personal Hygiene Practice Questionnaire.**

| Factor | Items | Factor Loadings (Rotated) | Uniqueness |
|--------|-------|--------------------------|------------|
| Factor 1 | Brushing teeth | 0.64 | 0.56 |
| | Bathing | 0.64 | 0.58 |
| | Wear clean clothes | 0.61 | 0.58 |
| | Change underwear | 0.59 | 0.56 |
| Factor 2 | Handwash with soap after toilet use | 0.53 | 0.62 |
| | Handwash after blowing nose | 0.45 | 0.69 |
| | Handwash after touching animal | 0.57 | 0.56 |
| | Handwash after touching private parts | 0.60 | 0.53 |
| Factor 3 | Changing bedsheet | 0.64 | 0.57 |
| | Changing pillow cover | 0.67 | 0.53 |
| | Dusting own room | 0.50 | 0.72 |
| | Mopping own room | 0.56 | 0.67 |

Bartlett's test of sphericity were 0.89 and χ² (153) = 8924.81, p < 0.001 respectively. The analysis retained three factors based on eigenvalues (>1), the scree plot and proportion of explained variance. Factor 1 is comprised of four items that are related to personal cleanliness practices (brushing teeth, bathing, wearing clean clothes and changing underwear), Factor 2 was associated with four items that were related to hand hygiene practice dimension, and Factor 3 was composed of another four items that reflected a dimension focused on keeping the surrounding clean (Fig 2). From Table 6, we found that factor 1 had a factor loading ranging from 0.59 to 0.64 and relatively low uniqueness value (0.56 to 0.58), factor 2 had loadings from 0.45 to 0.60 and uniqueness value ranging from 0.53 to 0.69, and factor 3, reflecting surrounding cleanliness with loadings between 0.50 and 0.67, and uniqueness value of 0.53 to 0.72. Overall, a factor loading ≥ 0.45 tells us that each item is well associated with the respective factors. Most of the item's uniqueness values fell below 0.70 in the EFA analysis which indicated that the extracted factors explained a considerable portion of variance in the personal hygiene questionnaire items. From these findings, our study can conclude that the questionnaire was a valid and reliable tool for assessing personal hygiene practice among university students.

## Discussion

The aim of our study was to assess the relationship between depression and personal hygiene practices among university students along with the validation of our newly developed personal hygiene questionnaire in order to make this self-rated questionnaire. In our study, we identified a statistically significant association between depression and personal hygiene practice. Depression risk among university students was high but higher in females than males. In terms of personal hygiene, female students maintain good personal hygiene practice compared with male. Along with gender, depression risk is associated with accommodation of students whereas personal hygiene practices are associated with level of study, accommodation, parental education and family income status.

The prevalence of depression risk in the present study among university students are higher than most recently published studies among students of Bangladesh. Though these studies used different scale of depression including the WHO-5 Well-Being Index (WHO-5) [17], and 9-item Patient Health Questionnaire (PHQ-9) [10,18], and Depression, Anxiety, and Stress Scale (DASS-42) scale [19]; the authors from these studies stated the prevalence of depression among university students ranging from 42% to 52%. These studies revealed a higher prevalence of depression level among female students compared with male [10,17–19]. However, a study conducted among first year university students reported a higher prevalence of depressive symptoms among male (50.4%) compared with female (49.6%) [2] inconsistent with the present study. Another study using DASS-21 scale among public university students reported a

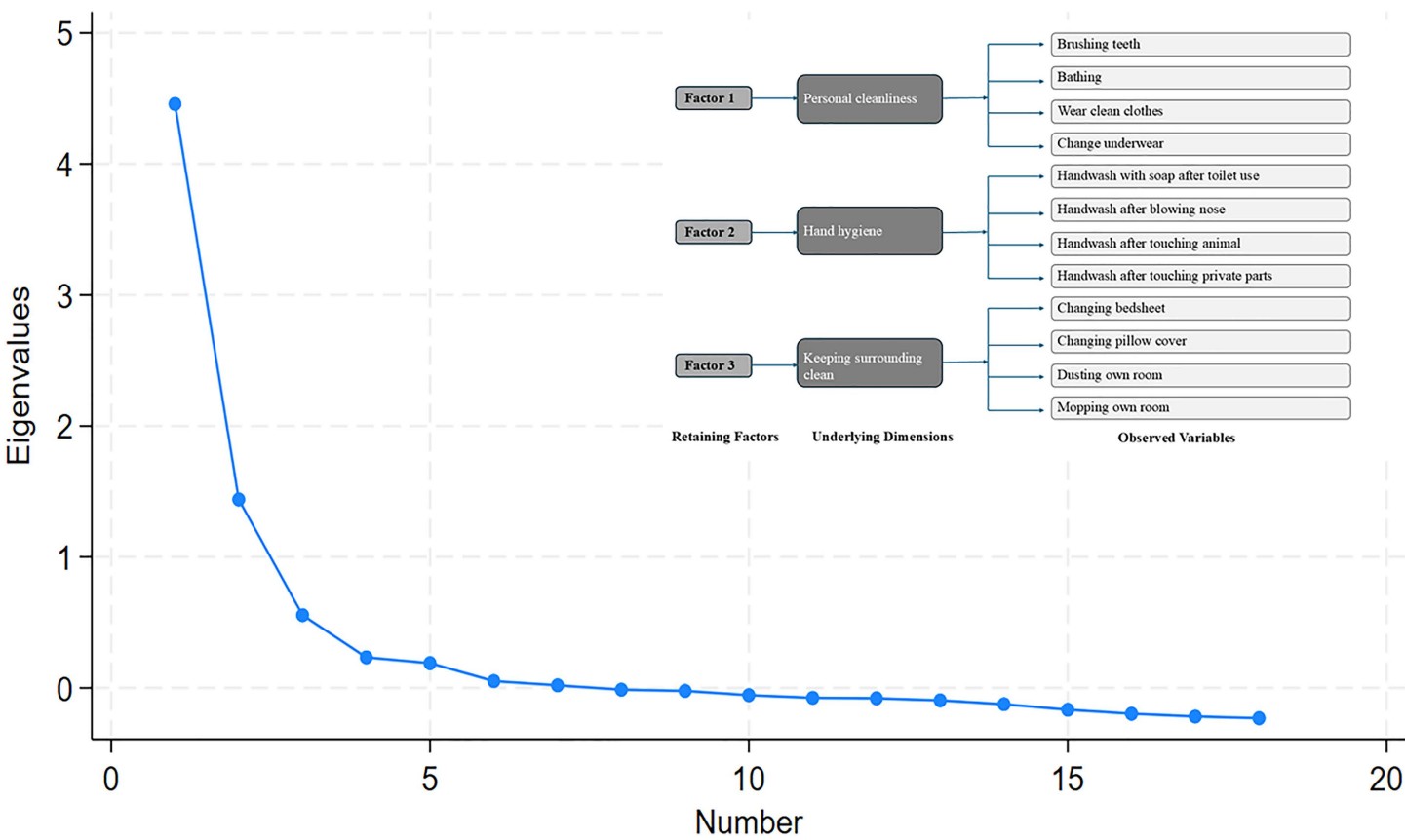

**Fig 2. Factor analysis by scree plot.**

non-significant higher prevalence of depression level among students who stayed at hall/mess [20]. The differences of depression prevalence regarding accommodation status in Hossain, Alam and Masum's study and our study could be due to inclusion of only public university students whereas we included both public and private university students.

The present study also evaluated the factors associated with personal hygiene among university students as there is a limited number of studies evaluated personal hygiene practices among university students in Bangladesh. Beyond the school-based WASH related articles, some studies evaluated the hand washing practices at university settings along with their knowledge and attitudes regarding this [21,22]. And to our best knowledge, no quantitative study was found to evaluate the personal hygiene practices considering other personal hygiene related variables as well as the impact of mental health on personal hygiene practices. This shortage of literature may limit the scope of comparing the findings of our study with others. The prevalence of practicing good personal hygiene was higher among female students than male in our study. Similar with present study, female students tended to greater use soap for hand washing within their college compared with men [5,23]. The absence of gender sensitivity at the contextual level serves as a significant barrier to the enhancement and promotion of sanitation and hygiene practices at the individual level. The factors that may relate to better personal hygiene practice among females are menstruation knowledge, family orientation, socio-cultural differences and physiological need for cleanliness [24]. The study noted that personal hygiene was significantly associated with socio-economic status. Consistent with present study, students belonging from middle or upper-middle income family had good personal hygiene practices as they have greater access to hygiene information (i.e., social media, newspaper, and other media exposure) [25,26]. Living place is considered as another reason for good personal hygiene among university students. Due to high economic

growth over the last two decades in Bangladesh, sanitation and hygiene facilities among school and university settings are increased rapidly. The demands of hygiene practices are also increased. To cover up the demands, the government of Bangladesh announced a program named "Sanitation for All by 2010" and all educational institutions are trying to comply with the national goals [27]. In our study, the percentage of maintaining good personal hygiene among university is quite high in both male and female, which may be due to practices of personal hygiene during the COVID-19 period.

Unhygienic personal behavior was associated with poor mental health. The present study revealed that the risk of depression among university students may reduce good personal hygiene practices. A recent study among health professionals found that depressive persons had low level of standards regarding personal hygiene and grooming [28] consistent with the present study. However, the study also observed excessive grooming and over-personal hygiene practices among depressed participants [28]. Low-economic status, lack of social activity and support and lack of proper vocational and academic opportunities have been linked with depression among individuals [29]. A recent scoping review revealed that depressed individuals were less likely to wash their hands. They did not have any guilt or did not have any intension to wash their hands with soap [30]. A GSHS analysis of four Southeast countries among middle school children found that students with one or more psychological distress were less likely to wash their hands after using toilet and less likely to brush their teeth [31]. Self-motivation and energy level are important factors that may influence hygiene and grooming practices [28]. Self-neglecting and early life trauma also linked with depression leading to difficulties in maintaining personal hygiene [28]. According to the definition of depression, individual's daily life activity at work or schools are impaired by depression [32]. However, another study conducted among children found inverse relationship between hand washing and depression [33]. Students reporting academic pressure emerged another predictive factor of depression which may lead to lower personal hygiene practices. The reasons behind this may be lack of time and proper management of academic activities to maintain personal hygiene. Education related to personal hygiene may also influence good personal hygiene practices. Engineering students have better personal hygiene practices compared with other educational background students. In contrast to our findings, students from health and life sciences background had better knowledge, attitude and practices regarding personal hygiene compared with engineering or other background students [34].

As we previously mentioned, literatures about personal hygiene practices among university is limited and no study yet carried out at Bangladesh settings, we had to develop and validate a new questionnaire tool to reduce the gap in literature for assessing the personal hygiene practices among students. EPA analysis suggested that this questionnaire has a correlated three factor structures. From the measurement of internal consistency and construct validity, this questionnaire has been demonstrated as a reliable measure for personal hygiene practices among university students.

## Limitations

This study has been limited by several factors. Firstly, the use of convenience sampling, which restricts the generalizability of the findings. Although we distributed the survey across diverse student networks to identify variation in university type, gender, and subject major, self-selection into the study means that certain groups may be over or underrepresented. Therefore, the results should be interpreted with caution when applying them to the wider university student population. Secondly, the pseudo-R-squared value for the regression model is much lower (0.089) indicating that the model can explain only a small portion of variance in the dependent variables although the findings are significant. Others unmeasured variables might influence our findings such as social welfare, personal care facility, social support, availability and accessibility to hygiene facilities, self-esteem, religious and cultural beliefs and practices and motivation. Thirdly, data were collected from universities in Dhaka, a central hub for higher education in Bangladesh, which may limit the generalizability to the broader university student population nationwide. Fourthly, the temporal stability of this questionnaire was not assessed. And the weight of some items in loadings 2 and 3 were considered low or moderate. And finally, we have used web-based data collection methods which may limit the number of participants as it can access to those who have internet. This may also introduce bias in sampling.

Despite these limitations, it is worth mentioning some strengths of our study. First, this is the first study in Bangladesh which evaluates the effect of depression on personal hygiene among university students. Secondly, we have used a newly developed and validate personal hygiene practices questionnaire with good reliability and validity score which can be used in future (after additional item inclusion) for assessing personal hygiene at university settings. Thirdly, we have included a diverse range of participations from different disciplines, study year, semester, residence to maximize the variation.

## Conclusion

This study provides a comprehensive analysis of the relationship between depression risk and personal hygiene practices among university students, revealing critical insights into the mental and physical health of this demographic. Gender and accommodation type were significant determinants; females displayed superior personal cleanliness habits, whilst students living in privately managed lodgings showed significant association between reduced depression and higher hygiene standards. The results of our study suggest that the association of mental health and personal hygiene among university students should be observed worldwide. These findings emphasize the necessity for educational institutions to focus mental health programs and hygiene instruction. More focus should be given to the cleanliness of accommodation area and facilities provided by university to address the problems related to depression and personal hygiene. Universities should also develop workshops focused on personal hygiene education, emphasizing its connection to mental well-being. Along with the educational programs, barriers related to social and institutional also need to be focused to find out the gaps in policy making by government.

Subsequent research should investigate the socio-economic determinants affecting personal cleanliness and mental health, as this study predominantly concentrated on demographic characteristics. Longitudinal studies were also suggested to evaluate the association of depression with personal hygiene across the lifespan from childhood to adulthood. This will help to understand the behavioral factors such as substances use, diet, and physical exercise that may have effect on the relationship between personal hygiene and depression. Exploring the impact of cultural beliefs and access to hygiene facilities will also provide a more nuanced understanding of these relationships. Addressing the intertwined issues of mental health and personal hygiene through targeted interventions can lead to improved health outcomes. By fostering a supportive and informed university environment, we can enhance the overall well-being of students, equipping them with the tools necessary for both academic success and personal health.

## Supporting information

**S1 Table. Comparison of Binary Logistic Regression of Personal Hygiene Coefficients to Assess Proportional Odds Assumption.**
(DOCX)

**S1 File. Zipped file of the Dataset.**
(XLSX)

## Acknowledgments

We are grateful to all the participants and appreciate the support of all enumerators. We are also thankful to all the faculty members who provided their insights into developing the questionnaire to assess personal hygiene practice.

## Author contributions

**Conceptualization:** Fouzia Akter.

**Data curation:** Fouzia Akter, Akibul Islam Chowdhury, Md. Nawal Sarwer.

**Formal analysis:** Fouzia Akter.

**Investigation:** Md. Nawal Sarwer.

**Methodology:** Fouzia Akter, Akibul Islam Chowdhury.

**Project administration:** Md. Nawal Sarwer.

**Resources:** Fouzia Akter, Akibul Islam Chowdhury.

**Software:** Fouzia Akter.

**Supervision:** Fouzia Akter, Akibul Islam Chowdhury.

**Validation:** Fouzia Akter.

**Visualization:** Fouzia Akter, Akibul Islam Chowdhury.

**Writing – original draft:** Fouzia Akter, Akibul Islam Chowdhury, Md. Nawal Sarwer.

**Writing – review & editing:** Fouzia Akter, Akibul Islam Chowdhury, Md. Nawal Sarwer.

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
