## [Editor Report · Decision Letter 0]

5 Jun 2025

Thank you for submitting your manuscript to PLOS ONE. After careful consideration, we feel that it has merit but does not fully meet PLOS ONE’s publication criteria as it currently stands. Therefore, we invite you to submit a revised version of the manuscript that addresses the points raised during the review process.

**1. Sampling & Data Collection**

**Unspecified sampling method** : The method used (e.g., convenience, snowball, stratified) is not clearly stated.**Generalized claim of diversity** : The phrase *"to ensure sample diversity"* lacks evidence—what dimensions of diversity were targeted (e.g., university type, gender, department)?**Non-standard date formatting** : Dates like "08/09/24" should follow academic standards (e.g., *September 8, 2024* ).**Missing response rate** : No mention of how many students were invited or the response rate, which affects generalizability.

**2. Measures (Sociodemographics, CES-D)**

**Vague terminology** : *"individual’s subject major"* should be replaced with *"students’ field of study"* .**Income categorization** : The quartile method for income is mentioned but not clearly explained—what ranges were used?**Missing translation details** : It is unclear whether the CES-D scale was used in English or translated into Bengali, and if so, whether it was culturally validated.

**3. Personal Hygiene Practice Questionnaire (PHPQ)**

**Mixed response scales** : Using both 3-point and 4-point Likert scales can confuse respondents and complicate scoring and analysis.**Lack of clear construct validation** : Although Cronbach’s alpha is provided, there’s no detail on *when* exploratory factor analysis (EFA) was conducted—before or after analysis?**Limited detail on development process** : While expert review and a pilot test were mentioned, there's no detailed breakdown of how items were selected, modified, or validated.**Scoring scheme complexity** : The scoring rules are somewhat convoluted and may introduce bias or inconsistency.

**4. Statistical Analysis**

**Repetition** : Cronbach’s alpha is mentioned multiple times unnecessarily.**Minor terminology error** : *"logistics regression"* should be *"logistic regression"* .**Assumption checks missing** : No mention of whether model assumptions (e.g., proportional odds for ordered logistic regression) were tested.**Weak sequencing** : Statistical methods could be better organized—from descriptive stats to inferential analysis.

**5. Language and Style**

**Grammatical issues** : Several grammatical and phrasing errors reduce clarity (e.g., plural agreement, missing articles).**Casual or unclear expressions** : Some sentences (e.g., “we got a value of...”) lack academic tone.**Lack of cohesion** : Transitions between sections are abrupt and need better integration.

**
Discussion:
**

·  **Lack of Clear Structure**

The discussion lacks a well-defined structure. Topics such as depression, personal hygiene, tool validation, and limitations are interwoven without clear thematic separation, making it harder to follow the argument logically.

·  **Redundancy and Awkward Phrasing**

Some sentences are repetitive or awkwardly constructed. For example:

"We also found a high prevalence of depression risk among male and female students but comparing with male, female students had a higher-level depression risk."

This could be rewritten more clearly as:

"Depression risk was high among all students, but higher among females than males."

·  **Superficial Analysis of Relationships**

While statistical associations are mentioned, the discussion lacks in-depth theoretical or psychological explanations for the observed relationships (e.g., how and why depression affects hygiene behaviors). Including relevant behavioral or mental health theories would strengthen the interpretation.

·  **Insufficient Exploration of Contradictory Findings**

The study acknowledges discrepancies with previous research (e.g., gender differences in depression), but does not delve deeply into possible reasons for these inconsistencies, such as cultural context, sample differences, or measurement tools.

·  **No Conceptual Framework Provided**

The study would benefit from presenting a conceptual model that visually or descriptively outlines the relationships between depression, hygiene practices, and demographic/socioeconomic factors.

·  **Lack of Concrete Recommendations**

Although implications are hinted at, the discussion does not clearly provide practical recommendations for universities, policymakers, or future researchers based on the study’s findings.

**
Conclusion:
**

 **Overstatement of Causality** :

“Reduced incidence of depression and higher hygiene standards”

The cross-sectional design does not allow causal claims. Phrases like “showed a reduced incidence of depression” or “leads to improved health outcomes” should be softened (e.g., “were associated with…”).

·  **Overly Idealistic Recommendations** :

While workshops and screenings are valuable, the text assumes feasibility and efficacy without considering potential barriers (e.g., resource constraints, stigma, institutional policy gaps). These should be acknowledged.

·  **Generalization to Broader Populations** :

Statements like “we can enhance the overall well-being of students” should be more cautious, given that the data are limited to a narrow student demographic in Dhaka.

·  **Lack of Linkage Between Findings and Interventions** :

The proposed interventions (e.g., peer support) are not clearly tied back to specific findings from this study. It would strengthen the conclusion to align each recommendation with a corresponding empirical result.

Samane Shirahmadi, PhD

Academic Editor

PLOS ONE

2. In the online submission form, you indicated that [All relevant data are within the manuscript. Dataset generated for this manuscript will be made available upon request from corresponding author.].

5. Please ensure that you refer to Figure 1-2 in your text as, if accepted, production will need this reference to link the reader to the figure.
---

## [Author Response · Author response to Decision Letter 1]

20 Jul 2025

Author Response to Reviewer

Manuscript ID: PONE-D-25-16896

Manuscript Title: Impact of depression on personal hygiene practices- A cross-sectional study among university students in Bangladesh

Reviewer Comment: Unspecified sampling method: The method used (e.g., convenience, snowball, stratified) is not clearly stated.

Authors Response: “We used a convenience sampling method to recruit participants for the study.”

Reviewer Comment: Generalized claim of diversity: The phrase "to ensure sample diversity" lacks evidence—what dimensions of diversity were targeted (e.g., university type, gender, department)?

Authors Response: “A structured questionnaire was distributed via a Google Form link to ensure accessibility for participants across various locations with the study nature, purpose and eligibility inclusion criteria for participation. The inclusion criteria required for the participants to be 1) a resident in Dhaka, 2) studying either public or private university, 3) studying in first/second/third/fourth year, 4) studying health or non-health background subject majors, 5) have access to computer or mobile with internet connection, 6) able to understand English and 7) written consent for participation ... Participation was voluntary, students who were interested accessed and completed the questionnaire. We shared invitations for participation in this study across both public and private university networks, student groups, and social media platforms.”

Reviewer Comment: Non-standard date formatting: Dates like "08/09/24" should follow academic standards (e.g., September 8, 2024).

Authors Response: “Recruitment of participants for data collection started on September 8, 2024, and ended on December 31, 2024.”

Reviewer Comment: Missing response rate: No mention of how many students were invited or the response rate, which affects generalizability.

Authors Response: Thank you for your valuable comment. We have corrected the information.

“The survey link was shared online on different platforms. Initially, 2,030 participants provided written informed consent online. After applying exclusion criteria for missing values and inconsistencies, 1913 respondents completed the entire survey, generating a response rate of 94.23%.”

Reviewer Comment: Vague terminology: "individual’s subject major" should be replaced with "students’ field of study".

Authors Response: The socio-demographic section contained data on age, gender, type of university, students’ field of study and their level of study, place of residence, parental education, and family income in Bangladeshi currency (BDT).

Reviewer Comment: Income categorization: The quartile method for income is mentioned but not clearly explained—what ranges were used?

Authors Response: “We used the quartile method based on participant’s self-reported monthly family income and divided the continuous income variable into four quartiles, where each quartile represents 25% of the sample. The cut-off value of each quartile was determined by Stata based on the distribution of family income variable in our dataset.”

Reviewer Comment: Missing translation details: It is unclear whether the CES-D scale was used in English or translated into Bengali, and if so, whether it was culturally validated.

Authors Response: We didn’t translate the CES-D scale in Bengali as one of our inclusion criteria for the participant was to be able to understand English. As the CES_D scale was not translated, further validation wasn’t required.

Reviewer Comment: Mixed response scales: Using both 3-point and 4-point Likert scales can confuse respondents and complicate scoring and analysis.

Authors Response: We used a combination of 3-point and 4-point Likert response scales in our questionnaire based on the nature of the items and expert input.

After developing the questionnaire, a pilot survey was conducted with 44 students to check whether they could easily understand the questions and response options. 93.18% of students didn’t find any difficulties in understanding the question and 86.36% of students had no issues with the options of the questions.

Reviewer Comment: Lack of clear construct validation: Although Cronbach’s alpha is provided, there’s no detail on when exploratory factor analysis (EFA) was conducted—before or after analysis?

Authors Response: To assess construct validity, we conducted Exploratory Factor Analysis (EFA) using principal factor extraction and varimax rotation before conducting the regression analysis.

Reviewer Comment: Limited detail on development process: While expert review and a pilot test were mentioned, there's no detailed breakdown of how items were selected, modified, or validated.

Authors Response: During questionnaire development, we initially created 21 items based on literature review. After consultations with faculty members from nutrition, pharmacy, and public health, we removed three items (questions on deodorant apply, wash own clothes, and use of tissue for nose picking) that were deemed redundant or less relevant, resulting in the final 18-item version. A pilot test among 44 students confirmed the clarity and appropriateness of items, and wording was refined based on student feedback.

Reviewer Comment: Scoring scheme complexity: The scoring rules are somewhat convoluted and may introduce bias or inconsistency.

Authors Response: We acknowledge the complexity in the use of both 3-point and 4-point Likert scales in the questionnaire. To maintain consistency and reduce potential bias in scoring, responses from both 3-point and 4-point Likert scales were recoded into a common scale of 0, 1, and 2.

Items on a three-point scale were scored as (i) "Always" = 2, "Sometimes" = 1, "Never" = 0; (ii) "Daily" = 2, "Weekly" = 1, "Monthly" = 0.

For the four-point scale, responses were coded as "Once a week/Once in 15 days" = 2, "Once a month" = 1, and "Once more than one month duration" = 0.

Reviewer Comment: Repetition: Cronbach’s alpha is mentioned multiple times unnecessarily.

Authors Response: Corrected in the result section.

Reviewer Comment: Minor terminology error: "logistics regression" should be "logistic regression".

Authors Response: Corrected

Reviewer Comment: Assumption checks missing: No mention of whether model assumptions (e.g., proportional odds for ordered logistic regression) were tested.

Authors Response: Thank you for the comment. We have done the assumption test for ordered logistic regression. We have added the table in the supplementary documents (S1 Table).

Reviewer Comment: Weak sequencing: Statistical methods could be better organized—from descriptive stats to inferential analysis.

Authors Response: We have organized the statistical methods from descriptive to inferential analysis in the revised version. Thank you.

Reviewer Comment: Grammatical issues: Several grammatical and phrasing errors reduce clarity (e.g., plural agreement, missing articles).

Authors Response: Corrected

Reviewer Comment: Casual or unclear expressions: Some sentences (e.g., “we got a value of...”) lack academic tone.

Authors Response: Corrected

Reviewer Comment: Lack of cohesion: Transitions between sections are abrupt and need better integration.

Authors Response: We have revised the manuscript and maintain the transitions for better integration.

Reviewer Comment: Lack of Clear Structure:

The discussion lacks a well-defined structure. Topics such as depression, personal hygiene, tool validation, and limitations are interwoven without clear thematic separation, making it harder to follow the argument logically.

Authors Response: Thank you very much, we have corrected the structure of the discussion section by following to follow the argument logically:

1. Describe the prevalence and associated factors of depression and personal hygiene

2. Then we describe the major findings of our study which is the association of depression with personal hygiene

3. And, then we discuss about the validation of our newly developed personal hygiene scale.

After that we discuss about the strengths and limitations of the study.

Reviewer Comment: Redundancy and Awkward Phrasing:

Some sentences are repetitive or awkwardly constructed. For example:

"We also found a high prevalence of depression risk among male and female students but comparing with male, female students had a higher-level depression risk."

This could be rewritten more clearly as:

"Depression risk was high among all students, but higher among females than males."

Authors Response: Thank you very much for the comments. We have corrected and paraphased the sentences that are repetitive and awkwardly constructed.

Reviewer Comment: Superficial Analysis of Relationships:

While statistical associations are mentioned, the discussion lacks in-depth theoretical or psychological explanations for the observed relationships (e.g., how and why depression affects hygiene behaviors). Including relevant behavioral or mental health theories would strengthen the interpretation.

Authors Response: Thank you for the comments.

We tried to write the discussion providing some theoritical explanation that may contribute to estabilishing the relationship between mental health and personal hygiene.

Reviewer Comment: Insufficient Exploration of Contradictory Findings:

The study acknowledges discrepancies with previous research (e.g., gender differences in depression), but does not delve deeply into possible reasons for these inconsistencies, such as cultural context, sample differences, or measurement tools.

Authors Response: Thank you very much for the comments. We have added some line describing the reasons behind the differences between the present studies with previous studies.

Reviewer Comment: No Conceptual Framework Provided:

The study would benefit from presenting a conceptual model that visually or descriptively outlines the relationships between depression, hygiene practices, and demographic/socioeconomic factors.

Authors Response: We have rewrite the discussion to provide a clear understanding and logical arguments and try to present a conceptual model to define the relationships between personal hygiene, depression and other socio-demographic characteristics.

Reviewer Comment: Lack of Concrete Recommendations

Although implications are hinted at, the discussion does not clearly provide practical recommendations for universities, policymakers, or future researchers based on the study’s findings.

Authors Response: Thank you very much for the comments. We have added some recommendations for the universities, policy makers and future researchers in the conclusion part.

Reviewer Comment: Overstatement of Causality:

“Reduced incidence of depression and higher hygiene standards”

The cross-sectional design does not allow causal claims. Phrases like “showed a reduced incidence of depression” or “leads to improved health outcomes” should be softened (e.g., “were associated with…”).

Authors Response: Thank you very much for the concern. We have correct the statement and remove the overstatement of causality from the conclusion part.

Reviewer Comment: Overly Idealistic Recommendations:

While workshops and screenings are valuable, the text assumes feasibility and efficacy without considering potential barriers (e.g., resource constraints, stigma, institutional policy gaps). These should be acknowledged.

Authors Response: Thanks to the reviewer. We have acknowledged the different barriers related to hygiene and depression at social and instutional level.

“Along with the educational programs, barriers related to social and institutional also need to be focused to find out the gaps in policy making by government.”

Reviewer Comment: Generalization to Broader Populations:

Statements like “we can enhance the overall well-being of students” should be more cautious, given that the data are limited to a narrow student demographic in Dhaka.

Authors Response: Thank you. We have removed the statements.

Reviewer Comment: Lack of Linkage Between Findings and Interventions:

The proposed interventions (e.g., peer support) are not clearly tied back to specific findings from this study. It would strengthen the conclusion to align each recommendation with a corresponding empirical result.

Authors Response: We have rewrite and added some lines to provide strengths in conclusion of the recent study.

---

## [Decision Letter · Decision Letter 1]

20 Aug 2025

Dear Dr. Fouzia Akter,

Thank you for submitting your manuscript to PLOS ONE. After careful consideration, we feel that it has merit but does not fully meet PLOS ONE’s publication criteria as it currently stands. Therefore, we invite you to submit a revised version of the manuscript that addresses the points raised during the review process.

We look forward to receiving your revised manuscript.

Kind regards,

Samane Shirahmadi, PhD

Academic Editor

PLOS ONE

Journal Requirements:

Reviewers' comments:

Reviewer's Responses to Questions

**Comments to the Author**

Reviewer #1: All comments have been addressed

Reviewer #2: (No Response)

2. Is the manuscript technically sound, and do the data support the conclusions?

Reviewer #1: Yes

Reviewer #2: Yes

3. Has the statistical analysis been performed appropriately and rigorously?

Reviewer #1: I Don't Know

Reviewer #2: Yes

4. Have the authors made all data underlying the findings in their manuscript fully available?

Reviewer #1: Yes

Reviewer #2: Yes

5. Is the manuscript presented in an intelligible fashion and written in standard English?

Reviewer #1: Yes

Reviewer #2: Yes

Reviewer #1: (No Response)

Reviewer #2: Dear Author

1. Ensure that all claims, especially those related to prevalence rates and contributing factors to depression, are up-to-date and supported by the most recent literature.

2.Literature Review:A more comprehensive literature review is needed. The introduction references several studies, but does not adequately synthesize findings or highlight gaps in the research. Discussing existing theories or models that relate to the relationship between hygiene and mental health would provide a solid theoretical framework.

3.Sampling Bias: The use of convenience sampling can lead to bias. The article should clarify any efforts made to recruit a representative sample, and discuss the implications this has for generalizability of findings.Overall, the methodology presented is thoughtfully developed with appropriate measures and analyses. Addressing the mentioned areas of improvement, particularly the discussion of sampling limitations, detailed statistical methods, and qualitative feedback from pilot testing, would enhance the clarity and robustness of the methodology section. 4.This will provide readers a clear understanding of the study's validity and reliability and potentially improve the study's impact in the field.

5.The discussion around cultural influences on hygiene practices and mental health in Bangladesh is limited. An exploration of local cultural factors that influence both personal hygiene and mental health would enrich the discussion.

6.The article does not sufficiently address potential confounding variables other than gender and accommodation type. Socioeconomic factors and lifestyle variables may confound results and should be critically analyzed.

7.Discussion of Findings: While the results highlight statistical significance, the discussion should focus more on the practical implications of these findings. What does this mean for university policy or mental health interventions?

8.Word Choice and Clarity:The use of the phrase "personal hygiene is crucial for maintaining overall health" should be evaluated; it can be too broad. A more precise statement focusing on mental health would be beneficial.

9.Minor grammatical issues should be rectified, such as ensuring consistency in the use of singular/plural forms throughout the text.

10.The references should be uniformly formatted according to the journal's style. They appear to be inconsistently presented throughout the document.

11.Abbreviations like PHPQ and CES-D should be defined at first use in the abstract or introduction, and then used consistently thereafter.

12.A dedicated limitations section could strengthen the paper. Discussing limitations related to the cross-sectional design, self-reported measures, and any potential biases would enhance transparency.

13. The conclusion repeats the findings from the results rather than synthesizing them into broader implications. It should provide a more rounded perspective on how the findings could impact policies or practices.

**Do you want your identity to be public for this peer review?** For information about this choice, including consent withdrawal, please see our Privacy Policy

Reviewer #1: No

Reviewer #2: **Yes: ** Parvin Cheraghi

---

## [Author Response · Author response to Decision Letter 2]

6 Sep 2025

Author’s Response to Reviewer’s Comments

Manuscript ID: PONE-D-25-16896R1

Manuscript Title: Impact of depression on personal hygiene practices- A cross-sectional study among university students in Bangladesh

Reviewer’s Comment: Ensure that all claims, especially those related to prevalence rates and contributing factors to depression, are up-to-date and supported by the most recent literature.

Author’s Response: Thank you very much. We have used recent literature (2020 to 2025) for reporting prevalence and other contributing factors related to depression and personal hygiene as much as possible.

Reviewer’s Comment: Literature Review: A more comprehensive literature review is needed. The introduction references several studies, but does not adequately synthesize findings or highlight gaps in the research. Discussing existing theories or models that relate to the relationship between hygiene and mental health would provide a solid theoretical framework.

Author’s Response: Thank you very much. We have highlighted the gaps found in different literatures and mentioned those in the introduction part to strengthen the study’s aim.

Reviewer’s Comment: Sampling Bias: The use of convenience sampling can lead to bias. The article should clarify any efforts made to recruit a representative sample, and discuss the implications this has for generalizability of findings. Overall, the methodology presented is thoughtfully developed with appropriate measures and analyses. Addressing the mentioned areas of improvement, particularly the discussion of sampling limitations, detailed statistical methods, and qualitative feedback from pilot testing, would enhance the clarity and robustness of the methodology section.

This will provide readers a clear understanding of the study's validity and reliability and potentially improve the study's impact in the field.

Author’s Response: We agree with the reviewer that the use of convenience sampling may introduce sampling bias and limit the generalizability of our findings. In our study, we tried to reduce this limitation by sharing the survey link in both public and private university networks, with the male and female student groups, and among students from both health and non-health backgrounds. Using this approach, we were able to identify variation across university types, gender, and academic disciplines. As the participants are self-selected into the study, the sample might not completely represent all university students in Dhaka. We have now added a clear statement in the methodology and discussion sections to acknowledge this limitation and to explain that our findings should be interpreted with caution when generalizing to the wider student population.

Reviewer’s Comment: The discussion around cultural influences on hygiene practices and mental health in Bangladesh is limited. An exploration of local cultural factors that influence both personal hygiene and mental health would enrich the discussion.

Author’s Response: Some lines about cultural influences have been added although there are limited literatures that evaluated the effect of cultural influences on personal hygiene or mental health. Thank you.

Reviewer’s Comment: The article does not sufficiently address potential confounding variables other than gender and accommodation type. Socioeconomic factors and lifestyle variables may confound results and should be critically analyzed.

Author’s Response: Thank you for the comment. We have adjusted for potential confounders (age, gender, university type, subject major, year of study, residence type, parental education, and family income quartile) in the regression analysis but wasn’t clearly mentioned in the write-up. Now, in the statistical analysis and result section of the revised manuscript, we have revised the write-up and clarified the confounders included in our regression analyses.

Reviewer’s Comment: Discussion of Findings: While the results highlight statistical significance, the discussion should focus more on the practical implications of these findings. What does this mean for university policy or mental health interventions?

Author’s Response: Thank you for the comments. We have added some practical implications line in the discussion section.

Reviewer’s Comment: Word Choice and Clarity: The use of the phrase "personal hygiene is crucial for maintaining overall health" should be evaluated; it can be too broad. A more precise statement focusing on mental health would be beneficial.

Author’s Response: We have read the overall manuscript again, and rewrite and changes the word where was necessary to clarify the sentences for reader.

Reviewer’s Comment: Minor grammatical issues should be rectified, such as ensuring consistency in the use of singular/plural forms throughout the text.

Author’s Response: We have read the manuscript and correct the grammatical mistakes and typos.

Reviewer’s Comment: The references should be uniformly formatted according to the journal's style. They appear to be inconsistently presented throughout the document.

Author’s Response: Thank you very much. We have rechecked every reference for uniformity and maintained the journal guideline.

Reviewer’s Comment: Abbreviations like PHPQ and CES-D should be defined at first use in the abstract or introduction, and then used consistently thereafter.

Author’s Response: Thank you for the comment. We have defined the abbreviation at first use and used them consistently thereafter.

Reviewer’s Comment: A dedicated limitations section could strengthen the paper. Discussing limitations related to the cross-sectional design, self-reported measures, and any potential biases would enhance transparency.

Author’s Response: We have included a dedicated limitation section.

Reviewer’s Comment: The conclusion repeats the findings from the results rather than synthesizing them into broader implications. It should provide a more rounded perspective on how the findings could impact policies or practices.

Author’s Response: We have organized and added some lines to provide broader implications of our study findings in the conclusion part.

---

## [Decision Letter · Decision Letter 2]

27 Oct 2025

Impact of depression on personal hygiene practices- A cross-sectional study among university students in Bangladesh

PONE-D-25-16896R2

Dear Dr. Fouzia Akter,

We’re pleased to inform you that your manuscript has been judged scientifically suitable for publication and will be formally accepted for publication once it meets all outstanding technical requirements.

Kind regards,

Samane Shirahmadi, PhD

Academic Editor

PLOS ONE

Reviewers' comments:

Reviewer's Responses to Questions

**Comments to the Author**

Reviewer #1: All comments have been addressed

Reviewer #2: All comments have been addressed

2. Is the manuscript technically sound, and do the data support the conclusions?

Reviewer #1: Yes

Reviewer #2: Yes

3. Has the statistical analysis been performed appropriately and rigorously?

Reviewer #1: Yes

Reviewer #2: Yes

4. Have the authors made all data underlying the findings in their manuscript fully available?

Reviewer #1: Yes

Reviewer #2: Yes

5. Is the manuscript presented in an intelligible fashion and written in standard English?

Reviewer #1: Yes

Reviewer #2: Yes

Reviewer #1: (No Response)

Reviewer #2: (No Response)

**Do you want your identity to be public for this peer review?** For information about this choice, including consent withdrawal, please see our Privacy Policy

Reviewer #1: **Yes: ** Salim Al-Huseini

Reviewer #2: **Yes: ** Parvin Cheraghi

---

## [Editor Report · Acceptance letter]

PONE-D-25-16896R2

PLOS ONE

Dear Dr. Akter,

I'm pleased to inform you that your manuscript has been deemed suitable for publication in PLOS ONE. Congratulations! Your manuscript is now being handed over to our production team.

Kind regards,

on behalf of

Dr. Samane Shirahmadi

Academic Editor

PLOS ONE